# The Capacity of APOB-Depleted Plasma in Inducing ATP-Binding Cassette A1/G1-Mediated Macrophage Cholesterol Efflux—But Not Gut Microbial-Derived Metabolites—Is Independently Associated with Mortality in Patients with ST-Segment Elevation Myocardial Infarction

**DOI:** 10.3390/biomedicines9101336

**Published:** 2021-09-27

**Authors:** Marina Canyelles, Álvaro García-Osuna, Alexandra Junza, Oscar Yanes, Núria Puig, Jordi Ordóñez-Llanos, Alessandro Sionis, Jordi Sans-Roselló, Aitor Alquézar-Arbé, David Santos, Noemi Rotllan, Josep Julve, Mireia Tondo, Joan Carles Escolà-Gil, Francisco Blanco-Vaca

**Affiliations:** 1Institut de Recerca de l’Hospital Santa Creu i Sant Pau, Institut d’Investigacions Biomèdiques, IIB Sant Pau, 08041 Barcelona, Spain; mcanyelles@santpau.cat (M.C.); npuigg@santpau.cat (N.P.); daymer11@hotmail.com (D.S.); NRotllanV@santpau.cat (N.R.); JJulve@santpau.cat (J.J.); 2Department of Clinical Biochemistry, Hospital de la Santa Creu i Sant Pau, IIB Sant Pau, 08041 Barcelona, Spain; agarciao@santpau.cat (Á.G.-O.); jordonez1952@gmail.com (J.O.-L.); 3CIBER de Diabetes y Enfermedades Metabólicas Asociadas (CIBERDEM), 28029 Madrid, Spain; alexandra.junza@urv.cat (A.J.); oscar.yanes@urv.cat (O.Y.); 4Department de Bioquímica i Biologia Molecular, Universitat Autònoma de Barcelona, 08041 Barcelona, Spain; 5Metabolomics Platform, Department of Electronic Engineering, Universitat Rovira i Virgili, 43204 Reus, Spain; 6Fundació per la Bioquímica i la Patologia Molecular, 08041 Barcelona, Spain; 7Servei de Cardiología, Hospital Santa Creu i Sant Pau, 08041 Barcelona, Spain; asionis@santpau.cat (A.S.); jordisansrosello@hotmail.com (J.S.-R.); 8CIBER de Enfermedades Cardiovasculares (CIBERCV), 28029 Madrid, Spain; 9Servei d’Urgències, Hospital Santa Creu i Sant Pau, 08041 Barcelona, Spain; aalquezar@santpau.cat

**Keywords:** HDL-mediated efflux, macrophages, trimethylamine N-oxide, trimethyllysine, myocardial infarction

## Abstract

Impaired HDL-mediated macrophage cholesterol efflux and higher circulating concentrations of trimethylamine N-oxide (TMAO) levels are independent risk factors for cardiovascular mortality. The TMAO precursors, γ-butyrobetaine (γBB) and Trimethyllysine (TML), have also been recently associated with cardiovascular death, but their interactions with HDL-mediated cholesterol efflux remain unclear. We aimed to determine the associations between APOB depleted plasma-mediated macrophage cholesterol efflux and plasma TMAO, γBB, and TML concentrations and explore their association with two-year follow-up mortality in patients with acute ST-elevation myocardial infarction (STEMI) and unstable angina (UA). Baseline and ATP-binding cassette transporter ABCA1 and ABCG1 (ABCA1/G1)-mediated macrophage cholesterol efflux to APOB-depleted plasma was decreased in patients with STEMI, and the latter was further impaired in those who died during follow-up. Moreover, the circulating concentrations of TMAO, γBB, and TML were higher in the deceased STEMI patients when compared with the STEMI survivors or UA patients. However, after statistical adjustment, only ABCA1/G1-mediated macrophage cholesterol efflux remained significantly associated with mortality. Furthermore, neither the TMAO, γBB, nor TML levels altered the HDL-mediated macrophage cholesterol efflux *in vitro.* We conclude that impaired ABCA1/G1-mediated macrophage cholesterol efflux is independently associated with mortality at follow-up in STEMI patients.

## 1. Introduction

The ability of high-density lipoprotein (HDL) particles to stimulate cholesterol efflux from macrophage foam cells, the first step of reverse cholesterol transport (RCT), is one major recognized HDL cardioprotective function [1]. Macrophage cholesterol efflux to HDL occurs via different mechanisms, termed simple aqueous diffusion, facilitated by scavenger receptor class B type I (SR-BI) and active transport induced by the transmembrane protein ATP-binding cassette transporter ABCA1 and ABCG1 (ABCA1/G1)-mediated pathways. The importance of HDL-mediated cholesterol efflux in atheroprotection emerged from a study reporting a strong inverse association between the ex vivo cholesterol efflux capacity of APOB-depleted serum, measured in cultured mouse macrophage foam cells, and the carotid intima-media thickness and likelihood of angiographical-defined coronary artery disease [2]. Despite another report found that higher cholesterol efflux was associated with an increased prospective risk of a composite cardiovascular endpoint of incident myocardial infarction (MI), stroke, or death [3], two subsequent studies confirmed an inverse association of HDL-mediated cholesterol efflux and incident coronary heart disease risk independent of HDL cholesterol (HDL-C) concentrations [4,5]. The HDL-mediated cholesterol efflux ability in mouse macrophage foam cells was further found inversely associated with cardiovascular mortality in patients with chronic coronary artery disease [6,7,8]. Moreover, a recent report showed that the serum cholesterol efflux capacity measured in human macrophages was also a strong predictor of all-cause mortality after a MI, but the study did not find any association with HDL-mediated efflux or ABCA1-dependent and SR-BI-mediated serum efflux capacities [9].

Trimethylamine N-oxide (TMAO), a gut microbial-derived metabolite of choline and L-carnitine, among other dietary precursors, has been associated with major adverse cardiovascular events [9,10,11]. Higher plasma TMAO concentrations have also been associated with higher mortality risk in patients with heart failure [12,13] and chronic kidney disease [14]. Two recent reports investigated the potential of TMAO concentrations in risk stratification after an MI. The two studies found that TMAO was a predictive biomarker of all-cause [15] and cardiovascular death [16], but only one of the studies found a role of TMAO as a predictor of reinfarction [15]. Finally, high levels of TMAO in plasma were also independently correlated with plaque rupture in patients with ST-segment elevation MI (STEMI) [17]. Beyond TMAO, γ-butyrobetaine (γBB), a major gut microbial metabolite produced from dietary L-carnitine, and its precursor Trimethyllysine (TML), have been associated with cardiovascular mortality in patients with carotid atherosclerosis [18]. TML and TMAO were also associated with major adverse cardiovascular events and all-cause mortality amongst patients presenting with acute coronary syndrome [11]. However, only TML (and not TMAO nor γ-BB) was found to predict acute MI in patients who underwent coronary angiography [19].

A potential link among TMAO, liver, and intestine cholesterol homeostasis and HDL function has been described [20]. TMAO may enhance foam cell formation by upregulating macrophage scavenger receptors, but, in experimental models, it has also been shown to deregulate enterohepatic cholesterol and bile acid metabolism and impairs macrophage RCT [21,22]. However, it has also been described that TMAO enhances macrophage cholesterol efflux, at least in part due to the upregulation of *Abca1* and *Abcg1* in peritoneal mouse macrophages [22]. Nonetheless, some studies have also reported TMAO downregulating *Abca1* gene expression in murine macrophages [23,24].

Altogether, although the prognostic value of macrophage cholesterol efflux and TMAO in predicting cardiovascular mortality in patients with MI has been reported, the association of both parameters together with other TMAO precursors has never been explored. Here, we aimed to evaluate the associations among HDL-mediated macrophage cholesterol efflux, TMAO, γBB, and TML with cardiovascular mortality in patients in the acute phase of STEMI and test whether the gut microbial-derived metabolites modulate the associations between HDL-mediated macrophage cholesterol efflux and mortality.

## 2. Experimental Section

### 2.1. Study Population and Data Collection

Blood samples from patients with STEMI (diagnosed according to the principles of the Universal Definition of Myocardial Infarction) [25,26] were obtained from a retrospective study that included 253 patients consecutively admitted to the Hospital de la Santa Creu i Sant Pau. The patients were followed for two years for cardiovascular adverse events, all-cause death, and hospital readmissions via telephone interview and/or electronic medical record review. The primary outcome of this study was cardiovascular death defined as death by cardiogenic shock, fatal MI, or heart failure. We selected a sub-sample of STEMI patients: 35 who died during admission or during follow-up and 36 who survived. In all STEMI patients, the Global Registry of Acute Coronary Events (GRACE 2.0) risk score was calculated [27]. A subgroup of 33 patients with suggested symptoms of MI but high-sensitive cardiac troponin T (hs-cTnT) concentrations below the cut-off used to define myocardial damage (14 ng/L) or not reaching an hourly increase >3.0 ng/L during three-hour serial sampling was diagnosed with unstable angina (UA) and used as the control group. All the subjects gave informed consent. The study was performed in accordance with the ethical principles set forth in the Declaration of Helsinki. UA and STEMI survivors were matched with the deceased STEMI group for age, sex, body mass index, arterial hypertension, and diabetes mellitus and smoking status.

### 2.2. Blood Samples and Biochemical Measurements

Blood samples were collected in EDTA anticoagulated tubes for the UA patients upon their admission and immediately before the angiographic procedure performed at the time of event in the STEMI patients. Plasma was obtained via centrifugation (10 min, at 10,000× *g*), hs-cTnT was measured in STAT-mode (Cobas e601, Roche Diagnostics, Basel, Switzerland), and aliquots were stored at −80 °C until analysis. Triglyceride, low-density lipoprotein cholesterol (LDL-C), and C-reactive protein (CRP) levels were measured enzymatically using commercial kits adapted for a COBAS 6000 autoanalyzer (Roche Diagnostics, Basel, Switzerland). Plasma HDL-C levels were measured after the precipitation of apolipoprotein (APO) B-containing lipoprotein particles with 0.44 mmol/L phosphotungstic acid (Merck, Darmstadt, Germany) and 20 mmol/L magnesium chloride (Sigma-Aldrich, Madrid, Spain). Plasma creatinine, to estimate the glomerular filtration rate (eGFR) (using the CKD-EPI formula), was analyzed with an Architect c16000 analyzer (Abbott Diagnostic, Chicago, IL, USA).

### 2.3. Ex Vivo and In Vitro Cholesterol Efflux Capacity

The ex vivo cellular cholesterol efflux was determined using TopFluor-cholesterol, a fluorescent cholesterol probe in which the cholesterol molecule is linked to a boron dipyrromethene difluoride (BODIPY) moiety (Avanti Polar Lipids, Alabaster, AL, USA). J774A.1 cells (7.5 × 10^4^/well) were seeded in 48-well plates and allowed to grow for 24 h in a DMEM-supplemented medium. Next, macrophages were labeled for 1 h in a high-glucose DMEM medium (Lonza, Waltham, MA, USA) containing 0.125 mmol/L total cholesterol, where the fluorescent cholesterol accounted for 20% of the total cholesterol complexed with 10 mmol/L methyl-β-cyclodextrin (Sigma-Aldrich, Madrid, Spain), as reported in [28]. The labeled cells were subsequently equilibrated for 18 h with DMEM containing 0.2% fatty-acid free BSA (Sigma-Aldrich, Madrid, Spain) and then incubated for 24 h with 1% APOB-depleted plasma in a Roswell Park Memorial Institute (RPMI) medium. The acyl-CoA cholesterol acyltransferase (ACAT) inhibitor, Sandoz 58-035 (5 µmol/L; Sigma Aldrich, Madrid, Spain), was present during the whole experimental procedure. This experiment was performed under baseline conditions and also under experimental settings mainly stimulating the concerted ABCA1/ABCG1-dependent efflux pathways by pre-treating the cells with 2 μmol/L of the liver X receptor agonist, T0901317 (Cayman Chemicals, Ann Arbor, MI, USA), during the equilibration process or, alternatively, under experimental settings that only stimulated the ABCA1-dependent pathway by pre-treating the cells with 0.3 mmol/L of cyclic adenosine monophosphate (cAMP, Sigma-Aldrich, Madrid, Spain). The fluorescence intensity was then measured in the medium using the microplate reader, Synergy HT (BioTek Instruments, Winooski, VT, USA), at λEx/Em = 485/530 nm. The cells were solubilized with 1% cholic acid and mixed on a plate shaker for 4 h at room temperature, and the fluorescence intensity was quantified. The cholesterol efflux capacity was calculated according to the formula: [media fluorescence/(media fluorescence + cells fluorescence)] × 100. All conditions were run in duplicate. Values were normalized using an APOB-depleted plasma pool within each efflux assay.

In the in vitro study, macrophages were pre-treated with 2 μmol/L of T0901317 and then incubated with 0, 5, 20, and 40 µM of TMAO, 0, 0.5, and 2 µM of γBB or 0, 0.5, 1, and 2 µM TML (Sigma-Aldrich, Madrid, Spain) for 24 h with 1% of and APOB-depleted plasma pool. The pool of plasma samples from normolipemic patients was obtained from the Clinical Laboratory of Hospital de la Santa Creu i Sant Pau. The cholesterol efflux capacity was determined as described above and expressed as % at 24h.

### 2.4. Plasma TMAO, γBB and TML Determinations

First, 25 µL of human plasma was mixed and vigorously vortexed for 20 s with 300 µL of acetonitrile: methanol: water (5:4:1; *v*:*v*:*v*) containing two internal standards (IS) for quantification. The internal standards used were d_3_-methylcarnitine (d_3_-MeCar) to quantify the γBB and TMAO concentrations and ^13^C_3_-TML to quantify the TML—both at 5 ppm. After 30 min of re-equilibration on ice, the samples were centrifuged for 10 min at 25,100× *g* and 4 °C, and the supernatant was transferred into the vial prior to the LC-MS analysis. To quantify the analytes in the plasma, matrix-matched calibration curves were prepared using a human plasma pool spiked with standards. The concentration ranges of the calibration curves were 0–250 µM, 0–25 µM and 0–20 µM for the TMAO, GBB and TML, respectively.

The extracts were analyzed using an ultra-high performance LC system coupled with a 6490 triple-quadrupole mass spectrometer (QqQ, Agilent Technologies, Santa Clara, CA, USA) with an electrospray ion source (LC-ESI-QqQ) working in positive mode. An ACQUITY UPLC BEH HILIC column (1.7 mm, 2.1 × 150 mm, Waters) and a gradient mobile phase consisting of water with 50 mM ammonium acetate (phase A) and acetonitrile (phase B) were used for chromatographic separation. The gradient was as follows: isocratic for 30 s at 75% B, from 0.5 to 2 min decreased to 65% B, from 2 to 2.1 min decreased to 45% B, from 2.1 to 3.9 min isocratic at 45% B, for 0.1 min raised to 75% B, and, finally, column equilibrated at 75% B until 5.5 min. The flow of the method was 0.6 mL/min. Then, 2 µL of plasma extract was injected in the LC system. The mass spectrometer parameters were as follows: drying and sheath gas temperatures of 280 °C and 400 °C, respectively; source and sheath gas flows of 20 and 12 L/min, respectively; nebulizer flow of 60 psi; capillary voltage of 2500 V; nozzle voltage of 500 V; and iFunnel HRF and LRF of 110 and 80 V, respectively. The QqQ worked in MRM mode using defined transitions. The transitions for TML, ^13^C_3_-TML (IS), GBB, TMAO, and d_3_-MeCar(IS) and the collision energy (CE(V)) were: TML 189→84(17), 189→130(30); ^13^C_3_-TML(IS) 192→84(21), 192→130(13); GBB 146→87(16), 146→60(12); TMAO 76→58(16), 76→59(8); d_3_-MeCar (IS) 165→63(16), 165→103(16).

### 2.5. Statistical Methods

Data is presented as the mean ± standard deviation (SD) for continuous variables and as frequencies and percentages for categorical variables. A chi-square test was used to compare the categorical data between groups. The normality of the data was analyzed using the Kolmogorov–Smirnov and D’Agostino and Pearson omnibus tests. A one-way analysis of variance (ANOVA) test was used to compare the continuous variables, and Tukey’s post-test was used for comparing differences among the groups. A Kruskal-Wallis test was used to compare continuous variables not following a Gaussian distribution, and Dunn’s post-test was used for comparing differences among the groups. Correlations between variables were analyzed using Pearson’s correlation analysis. Multivariate analysis of covariance (ANCOVA) was used to explore the associations with mortality, adjusting for potential confounders. The statistical software, IBM SPSS Statistics v23, and GraphPad Prism 6.0 software (GraphPad, San Diego, CA, USA) were used to perform all statistical analyses. A *p*-value < 0.05 was considered to be statistically significant.

## 3. Results

### 3.1. Study Subjects

The baseline clinical and plasma biochemical characteristics of patients admitted with UA and those with STEMI who survived or died in a two-year follow-up period are shown in Table 1. The triglyceride concentrations did not differ among the three patient groups. HDL-C concentrations were significantly lower in patients with STEMI compared to those with UA but not different between the survivors and deceased STEMI patients. The deceased STEMI patients showed significantly lower LDL-C when compared with the survivors, and the latter presented higher LDL-C concentrations than the UA patients. Logarithmically transformed plasma CRP values were higher in patients with STEMI compared with those in the UA group, and this value was higher in the deceased STEMI patients when compared with the survivors (Table 1). The eGFR was also significantly lower in the deceased STEMI patients when compared with the STEMI survivors and those admitted with UA (Table 1). Also, the GRACE risk score and hs-cTnT were higher in the STEMI deceased patients when compared with the STEMI survivors (Table 1). The percentage of STEMI patients treated with statins was 38% for survivors and 44% for deceased (*p* = 0.811).

### 3.2. ABCA1 and ABCG1-Mediated Macrophage Cholesterol Efflux to APOB-Fepleted Serum Ex Vivo Is Down Regulated in Deceased STEMI Patients

The ex vivo ability of APOB-depleted plasmas in inducing macrophage cholesterol efflux was evaluated in all the groups under baseline conditions and experimental settings stimulating only the ABCA1 or both the ABCA1/G1 pathways. The baseline macrophage cholesterol efflux was significantly lower in the deceased STEMI patients when compared with the STEMI survivors and those with UA (Figure 1A). The ABCA1-mediated macrophage cholesterol efflux was also decreased in the patients who died when compared with the UA group (Appendix A). This change was also observed after stimulating both the ABCA1/G1-dependent efflux pathways, but, in this case, the macrophage cholesterol efflux was also more decreased in the STEMI survivors when compared with the UA patients (Figure 1B).

### 3.3. Circulating Levels of TMAO, γBB, and TML Are Increased in STEMI Patients Who Died during Follow-Up

Plasma concentrations of TMAO, γBB and TML were assessed in the same samples that APOB-depleted plasma-mediated cholesterol efflux was determined. The levels of these metabolites were significantly higher in the deceased STEMI patients compared to the STEMI survivors and UA patients (Figure 2).

### 3.4. ABCA1/G1-Mediated Macrophage Cholesterol Efflux Is Independently Associated with Mortality

We further evaluated the association between APOB-depleted plasma-mediated macrophage cholesterol efflux and mortality in these patients after adjusting for the main covariates that were significantly correlated with macrophage cholesterol efflux, including HDL-C, TMAO, and eGFR (all with *p* < 0.05). The association between the ABCA1/G1-mediated macrophage cholesterol efflux and death remained significant after adjusting for these parameters in (a) of (Table 2). However, both the baseline and ABCA1-mediated macrophage cholesterol efflux were not significantly associated with death following adjustment for these factors (Appendix A). Furthermore, the ABCA1/G1-mediated macrophage cholesterol efflux association with death also remained significant when TML (adjusted R-squared = 0.469; *p* = 0.009) or γBB was included in the analyses (adjusted R-squared = 0.483; *p* = 0.009). Next, we further adjusted for other covariates that correlated with the ABCA1/G1-mediated macrophage cholesterol efflux (*p* < 0.05) which strongly predict STEMI mortality, such as hs-cTnT, GRACE score, HDL-C and CRP. The association of the ABCA1/G1-mediated macrophage cholesterol efflux with death still remained significant after adjustment in (b) of (Table 2).

Also, we evaluated the association between gut microbial-derived metabolites and mortality in these patients after adjusting for the main determinant of these metabolites, eGFR, and also for the main parameter of interest, the ABCA1/G1-mediated macrophage cholesterol efflux. The association between TMAO (adjusted R-squared = 0.099; *p* = 0.088) or γBB (adjusted R-squared = 0.104; *p* = 0.123) with death did not remain significant after adjusting for these parameters. In contrast, TML retained its independent association with death after adjustment for these parameters in (a) of (Table 3). However, the independent association between TML and mortality did not persist after adjustment for hs-cTnT, GRACE score and CRP in (b) of (Table 3).

### 3.5. TMAO, γBB, and TML Did Not Affect APOB-Depleted Plasma-Mediated Macrophage Cholesterol Efflux In Vitro

Finally, we determined the effects of TMAO, γBB, and TML on the capacity of an APOB-depleted plasma pool to induce macrophage cholesterol efflux *in vitro*. For this purpose, the macrophages were incubated with increasing concentrations of all metabolites during the efflux period of 24 h. Importantly, TMAO, γBB and TML did not modify the ABCA1/G1-mediated macrophage cholesterol efflux to an APOB-depleted plasma pool (Figure 3A–C).

## 4. Discussion

Macrophage cholesterol efflux is largely induced by HDL particles. A recent meta-analysis showed that the macrophage cholesterol efflux capacity was inversely associated with the risk of cardiovascular disease and mortality, although the authors also admitted the heterogeneity among the included studies and evidence of publication bias [29]. Part of this heterogeneity could be explained by the methodology used to isolate the HDL particles; many studies have analyzed cholesterol efflux after the chemical precipitation of APOB-containing lipoproteins to isolate HDL particles from serum or plasma, while others used different approaches. Three independent studies showed that the impaired cholesterol efflux capacity of APOB-depleted serum or plasma from mouse macrophages was associated with higher cardiovascular mortality in patients with chronic coronary artery disease [6,7,8]. In these studies, mouse J774.A1 macrophages were incubated with cAMP to upregulate ABCA1, thereby suggesting the major contribution of a dysfunctional ABCA1-mediated pathway in this cellular model. However, a recent report failed to find any significant association between ABCA1 or SR-BI-mediated cholesterol efflux and the cardiovascular mortality of MI patients, even though it did show that their serum cholesterol efflux capacity was a strong predictor of mortality after a MI [9]. It is noteworthy that this report used human THP-1 macrophages pretreated with acetyl-LDL, which stimulates both ABCA1/G1-mediated efflux [9]. Our work adds to current knowledge that the impaired ABCA1/G1-mediated cholesterol efflux capacity from mouse J774.A1 macrophages to HDL is independently associated with mortality in STEMI patients. Our results reinforce the importance of measuring different components of macrophage cholesterol efflux capacity, integrating both the ABCA1-dependent pathway, which facilitates cholesterol release to lipid-free apoA-I/preβ-HDL, and the ABCG1-dependent efflux to mature α-migrating HDL particles. These assays better mimic the two predominant pathways of cholesterol-loaded macrophages converted into foam cells in atherosclerotic lesions [30].

Our analyses revealed that patients with STEMI who had higher plasma TMAO, γBB, and TML were more likely to experience subsequent death. However, the associations between TMAO or γBB and death were attenuated when further adjusted for other factors. Despite several reports showing that TMAO independently predicted future cardiovascular adverse events [11,31], not all reports have found independent associations between TMAO levels and cardiovascular outcomes [32,33,34]. The strong influence of renal function on TMAO levels could explain, at least in part, the inconsistency of TMAO in predicting adverse cardiovascular outcomes [32,33].

TML can be obtained from diet but also from endogenous production via the methylation of lysine residues in histones and, most likely, under the proteolytic degradation of other proteins; furthermore, TML is a poor precursor of TMAO [31]. Of note, the association of TML and death remained significant after adjusting for the main gut microbial-derived metabolite determinants in our study, but it was lost after adjusting for major covariates predicting STEMI mortality [35]. However, other studies have shown TML as an independent prognostic marker of major adverse cardiovascular events and mortality risk [11,19,31]. Our results do not necessarily contradict those obtained in patients with suspected stable angina pectoris [19], in subjects without evidence of acute coronary syndrome who underwent elective diagnostic coronary angiography [31], or those who had UA or adjudicated acute coronary syndromes [11]. Therefore, TML levels could provide insight regarding cardiovascular risk assessment but, at least among patients with prior STEMI, were found to not offer additional information for predicting mortality. Overall, our findings do not support the role of TML as a pure specific marker for STEMI mortality.

Previous reports have shown divergent results regarding TMAO-mediated effects on macrophage transporters involved in cholesterol efflux, specifically indicating enhancing [22], neutral [24], or downregulating effects [23]. The present study goes further and demonstrates that TMAO, γBB, and TML, at physiological levels, did not produce significant effects on macrophage cholesterol efflux, indicating that the first step in the RCT process was not affected by these three compounds. Taken together, our data does not support any implication of TMAO, γBB, and TML in modulating the association between ABCA1/G1-mediated macrophage cholesterol efflux and STEMI mortality.

The present study has some limitations. The number of patients in our cohort was relatively modest, but it was compensated for the fact that we selected homogeneous and well-characterized patient groups, thus providing enough power to conduct our analyses.

## 5. Conclusions

We demonstrate that reduced ABCA1/G1-mediated macrophage cholesterol efflux is independently associated with mortality in STEMI patients. Consistent with the gradual severity of acute coronary syndrome, STEMI patients who survived presented higher ABCA1/G1-mediated macrophage cholesterol efflux than those who died. TMAO, γBB, and TML did not affect either macrophage cholesterol efflux to APOB-depleted plasma or the association between ABCA1/G1-mediated macrophage cholesterol efflux and mortality. These results motivate further research on therapeutic strategies aiming to improve the ABCA1/G1-mediated macrophage cholesterol efflux capacity.

## Figures and Tables

**Figure 1 biomedicines-09-01336-f001:**
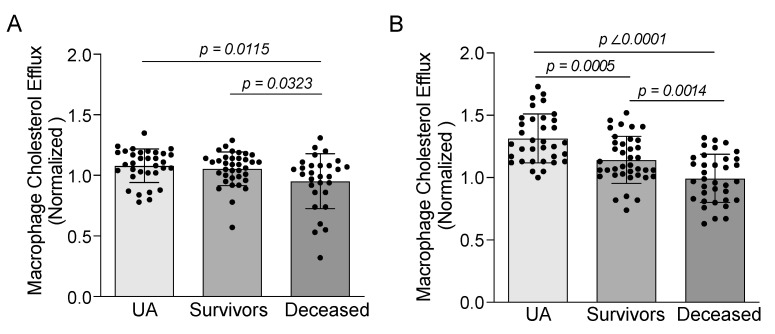
Macrophage cholesterol efflux is impaired in STEMI patients who died within a two-year follow-up period under baseline conditions (ANOVA *p* value = 0.0096) (**A**) and after activation of macrophage ABCA1/G1-dependent pathways by the LXR agonist, T090137 (ANOVA *p* value < 0.0001) (**B**). Values are represented as mean ± SD for UA (*n* = 33), STEMI survivors (*n* = 36) and deceased STEMI (*n* = 35).

**Figure 2 biomedicines-09-01336-f002:**
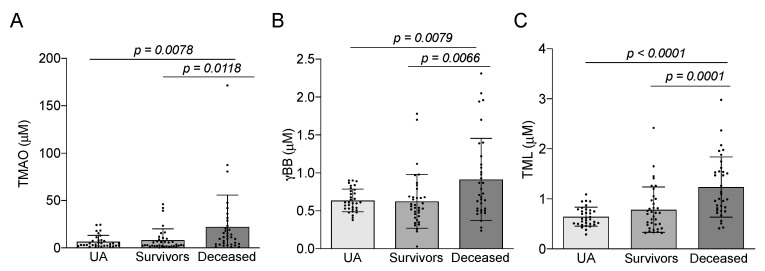
Plasma TMAO (ANOVA *p* value = 0.0040) (**A**); γBB (ANOVA *p* value = 0.0029) (**B**) and TML (ANOVA *p* value < 0.0001) (**C**) levels were higher in deceased STEMI patients than in the other two groups. Values are represented as mean ± SD in UA patients (*n* = 33), STEMI survivors (*n* = 36), and deceased STEMI patients (*n* = 35).

**Figure 3 biomedicines-09-01336-f003:**
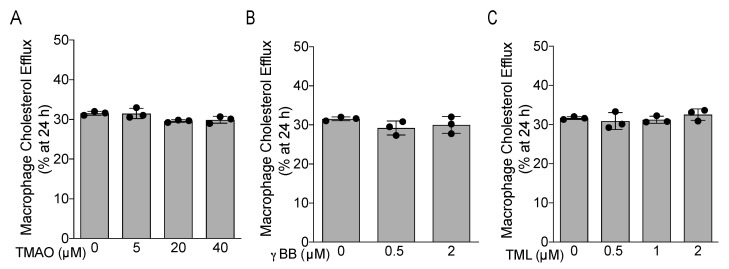
Effects of TMAO, γBB, and TML on ABCA1/G1-mediated macrophage cholesterol efflux to APOB-depleted plasma. The macrophage cholesterol efflux capacity was expressed as % of fluorescence released at 24 h. TMAO (**A**), γBB (**B**), and TML (**C**) levels did not affect the macrophage cholesterol efflux to the APOB-depleted plasma under the conditions stimulating the ABCA1/G1-dependent pathways. Kruskal-Wallis and Dunn’s post-test did not reveal significant differences among the groups incubated with the gut microbial-derived metabolites versus the control group without TMAO, γBB, and TML. Three independent experiments were performed for each condition.

**Table 1 biomedicines-09-01336-t001:** Clinical and biochemical parameters of the studied patients at admission.

	UA *n* = 33	STEMI Survivors *n* = 36	STEMI Deceased *n* = 35	*p*-Value
Age (years)	71.97 ± 3.45	72.06 ± 7.3	73.41 ± 9.86	0.6656
Sex (M/F)	17/16	20/16	19/15	0.923
Body mass index (Kg/m^2^)	26.38 ± 2.86	26.66 ± 2.06	27.07 ± 4.07	0.6774
Hypertension (%)	56	72	71	0.3525
Diabetes mellitus (%)	21	25	41	0.1903
Smoking (%)	17	25	35	0.0859
Triglycerides (mmol/L)	1.07 ± 0.47	0.97 ± 0.4	1.12 ± 0.73	0.5012
HDL-C (mmol/L)	1.63 ± 0.44	1.33 ± 0.34 **	1.19 ± 0.31 ****	<0.0001
LDL-C (mmol/L)	2.21 ± 0.64	2.81 ± 1.01 **	2.12 ± 0.89 ††	0.0022
Log_10_ C-reactive protein (mg/L)	3.30 ± 0.63	3.78 ± 0.71 *	4.42 ± 0.87 **** ††	<0.0001
eGFR (mL/min/1.73 m^2^)	77.74 ± 13.94	69.66 ± 18.21	51.11 ± 20.50 **** ††††	<0.0001
GRACE 2.0 risk score	ND	188.5 ± 44.23	251.6 ± 44.35 ††††	<0.0001
Log_10_ hs-cTnT (ng/L)	3.90 ± 0.17	5.36 ± 0.79 ****	6.06 ± 0.72 †††† ****	<0.0001

UA: unstable angina; STEMI: ST-segment elevation myocardial infarction; HDL-C and LDL-C: high- and low-density lipoprotein cholesterol; eGFR: estimated glomerular filtration rate by CKD-EPI equation; GRACE 2.0: Global Registry of Acute Coronary Events. ND, non-determined. Results are presented as mean ± standard deviation (SD). * *p* < 0.05 ** *p* < 0.01 **** *p* < 0.0001 vs. UA†† *p* < 0.01 †††† *p* < 0.0001 vs. STEMI survivors.

**Table 2 biomedicines-09-01336-t002:** Analysis of covariance (ANCOVA) of ABCA1/G1-mediated macrophage cholesterol efflux at 24 h adjusted for (a) HDL-C, eGFR and TMAO that significantly correlated with the dependent variable or (b) hs-cTnT, GRACE score, HDL-C and CRP, factors strongly associated with mortality.

**(a)**					
**Source**	**SS**	**Df**	**Mean Square**	**F**	***p* Value**
Corrected model	2.610 *	5	0.522	18.403	0.000
Intercept	2.287	1	2.287	80.636	0.000
HDL-C	0.776	1	0.776	27.371	0.000
eGFR	0.043	1	0.043	1.509	0.222
TMAO	0.000	1	0.000	0.011	0.917
Death	0.331	2	0.166	5.838	0.004
Error	2.666	94	0.028		
Total	139.441	100			
Corrected total	5.275	99			
* R-squared = 0.495 (adjusted R-squared = 0.468)					
**(b)**					
**Source**	**SS**	**Df**	**Mean Square**	**F**	***p* Value**
Corrected model	0.755 *	5	0.151	4.652	0.001
Intercept	0.119	1	0.119	3.674	0.060
hs-cTnT	0.034	1	0.034	1.054	0.309
GRACE score	0.007	1	0.007	0.213	0.646
HDL-C	0.440	1	0.440	13.553	0.000
CRP	0.000	1	0.000	0.008	0.931
Death	0.188	1	0.188	5.804	0.019
Error	2.012	62	0.032		
Total	81.888	68			
Corrected total	2.767	67			
* R-squared = 0.114 (adjusted R-squared = 0.058)					

**Table 3 biomedicines-09-01336-t003:** Analysis of ANCOVA of circulating TML levels adjusted for (a) eGFR and normalized ABCA1/G1-mediated efflux or (b) hs-cTnT, GRACE score and CRP.

**(a)**					
**Source**	**SS**	**df**	**Mean Square**	**F**	***p* Value**
Corrected model	7.423 *	4	1.856	10.054	0.000
Intercept	4.155	1	4.155	22.512	0.000
eGFR	1.415	1	1.415	7.665	0.007
Normalized ABCA1/G1-mediated efflux	0.002	1	0.002	0.011	0.917
Death	1.934	2	0.967	5.239	0.007
Error	17.535	95	0.185		
Total	103.990	100			
Corrected total	24.958	99			
* R Squared = 0.297 (Adjusted R Squared = 0.268)					
**(b)**					
**Source**	**SS**	**df**	**Mean Square**	**F**	***p* Value**
Corrected model	5.948 *	4	1.487	5.554	0.001
Intercept	0.051	1	0.051	0.191	0.663
hs-cTnT	0.005	1	0.005	0.018	0.894
GRACE score	1.733	1	1.733	6.471	0.013
CRP	1.455 × 10^−5^	1	1.455 × 10^−5^	0.000	0.994
Death	0.596	1	0.596	2.224	0.141
Error	17.137	64	0.268		
Total	94.710	69			
Corrected total	23.086	68			
* R-squared = 0.258 (adjusted R-squared = 0.211)					

## Data Availability

The data that support the findings of this study will be available to other researchers upon reasonable request.

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
