# Peer review of "The Capacity of APOB-Depleted Plasma in Inducing ATP-Binding Cassette A1/G1-Mediated Macrophage Cholesterol Efflux—But Not Gut Microbial-Derived Metabolites—Is Independently Associated with Mortality in Patients with ST-Segment Elevation Myocardial Infarction"

_biomedicines, 2021, doi:10.3390/biomedicines9101336_

Round 1

Reviewer 1 Report

The authors showed that the reduction of macrophage cholesterol efflux via ABC transporters (ABCA1, ABCG1) is independently associated with mortality in STEMI patients. Patients with STEMI who survived had greater eflux cholesterol from macrophages compared with those who died. Moreover, no effect of N-trimethylamine, gamma-butyrbetaine and trimethyl lysin on cellular cholesterol efflux into ApoB depleted plasma was observed. The relationship between cholesterol efflux and mortality in the study group was also not confirmed.
The research is interesting. Although the hypotheses have not been confirmed, the results of the study provide completely new information on macrophage cholesterol efflux and will be helpful in continuing research in this field.

Author Response

The authors showed that the reduction of macrophage cholesterol efflux via ABC transporters (ABCA1, ABCG1) is independently associated with mortality in STEMI patients. Patients with STEMI who survived had greater eflux cholesterol from macrophages compared with those who died. Moreover, no effect of N-trimethylamine, gamma-butyrbetaine and trimethyl lysin on cellular cholesterol efflux into ApoB depleted plasma was observed. The relationship between cholesterol efflux and mortality in the study group was also not confirmed.
The research is interesting. Although the hypotheses have not been confirmed, the results of the study provide completely new information on macrophage cholesterol efflux and will be helpful in continuing research in this field.

We acknowledge the reviewer for the comments. The reviewer is right, although we did not confirm all the initial hypothesis, we have demonstrated that ABCA1/G1-mediated cholesterol efflux  contributes independently to  predicting mortality in STEMI patients. We agree that confirmation in other patient cohorts will be needed to give further strength to the concept emerging from our and previous studies.

Reviewer 2 Report

The authors showed that the capacity of APOB-depleted plasma in ABCA1/ABCG1-mediated macrophage cholesterol efflux is independently associated with mortality at follow-up in patients with stem MI. The plasma levels of TMAO, γBB, and TML were higher in the deceased STEMI patients than STEMI survivors or UA patients, but they did not alter HDL-mediated macrophage cholesterol efflux. This topic is interesting, but there are some points to be considered.

2.2 Blood samples and biochemical measurements

Blood samples in patients with STEMI were obtained in the hemodynamic laboratory prior to a coronary angiography. Was the hemodynamic laboratory prior to a coronary angiography performed during follow-up or at the time of the event?

2.4. Plasma TMAO, γBB and TML determinations

25 µL of mouse/human plasma・・・, In which experiments did the authors use mouse blood?

2.5. Cell Culture and Treatment

Please describe the origin of the APOB-depleted plasma pool.

3.1 Study subjects

The deceased STEMI patients showed significantly lower LDL-C when compared with the survivors,・・・Please examine whether or not patients in each group were treated with lipid-lowering drugs.

Table 2,3

Please add a description for A and B in Table 2 and 3 and a in Table 2. The reviewer did not understand the difference between 1 and 2 in Df of death.

How did the authors choose the covariates? Because the authors described that UA and STEMI survivors were matched with the deceased STEMI group for age, sex, BMI, arterial hypertension, and DM and smoking status, the reviewer thinks age and BMI were not covariates. Isn’t LDL-C also a covariable?

3.4 ABCA1/G1-mediated macrophage cholesterol efflux is ~.

Is statistical analysis performed using only STEMI patients in this paragraph? The data for which group is difficult to understand. There is not enough explanation of the data thorough the manuscript.

Figure 3

There is a large variation in Figure 3A-C. The authors should increase the number. Please add the explanation of “macrophage cholesterol efflux (% at 24h)”.

Author Response

The authors showed that the capacity of APOB-depleted plasma in ABCA1/ABCG1-mediated macrophage cholesterol efflux is independently associated with mortality at follow-up in patients with stem MI. The plasma levels of TMAO, γBB, and TML were higher in the deceased STEMI patients than STEMI survivors or UA patients, but they did not alter HDL-mediated macrophage cholesterol efflux. This topic is interesting, but there are some points to be considered.

We acknowledge the reviewer for the comments and suggestions which have helped us to significantly improve the manuscript. Our responses are presented below, and the changes have been highlighted in the revised version of the manuscript.

 2.2 Blood samples and biochemical measurements

Blood samples in patients with STEMI were obtained in the hemodynamic laboratory prior to a coronary angiography. Was the hemodynamic laboratory prior to a coronary angiography performed during follow-up or at the time of the event?

All blood samples of the study were obtained in the acute phase of STEMI. In our health system, once STEMI patients are identified, they are transferred without any intermediate step to the hemodynamic laboratories of reference hospitals for coronary catheterization, with both diagnostic and therapeutic aims. At the hemodynamic units, the clinical workout includes a blood drawing from a peripheral venous line for clinical laboratory testing immediately before the procedure. The leftover of such samples were used in the present study. We have now included that “Blood samples were obtained immediately before the angiographic procedure performed at the time of event.” to clarify this point in the new version of the manuscript.

 2.4. Plasma TMAO, γBB and TML determinations

25 µL of mouse/human plasma・・・, In which experiments did the authors use mouse blood?

It was an error. We have remove “mouse” from the text.

2.5. Cell Culture and Treatment

Please describe the origin of the APOB-depleted plasma pool.

A pool of plasma samples from normolipemic patients was obtained from the Clinical Laboratory of Hospital de la Santa Creu i Sant Pau. This pool was later treated with 0.44 mmol/L phosphotungstic acid (Merck, Darmstadt, Germany) and 20 mmol/L magnesium chloride (Sigma-Aldrich, Madrid, Spain) to precipitate apoB-containing lipoproteins. We have now included the origin of APOB-depleted plasma pool in the section 2.3.

3.1 Study subjects

The deceased STEMI patients showed significantly lower LDL-C when compared with the survivors,・・・Please examine whether or not patients in each group were treated with lipid-lowering drugs.

The percentage of STEMI patients treated with statins was 38 % for survivors and 44% for deceased. No statistically differences were found between two groups (p = 0.811). We have included this point in the new version of the manuscript.

Table 2,3

Please add a description for A and B in Table 2 and 3 and a in Table 2. The reviewer did not understand the difference between 1 and 2 in Df of death.

We have added a brief description for A and B in the tables and defined a (now as *). The different number in Df was due to the fact that this calculation takes into account the number of groups and covariates used in the analysis (i.e. 3 groups and 4 covariates in Table 2B, except for GRACE 2.0 risk scores that they were determined only in STEMI patients). Also, we have clarified that the sign provides the statistical significance value (i.e., p-value).

.

How did the authors choose the covariates? Because the authors described that UA and STEMI survivors were matched with the deceased STEMI group for age, sex, BMI, arterial hypertension, and DM and smoking status, the reviewer thinks age and BMI were not covariates. Isn’t LDL-C also a covariable?

The covariates in the Tables 2 and 3 (A) were originally chosen due to their significant correlation with the dependent variable. Next, we also used other covariates in B because of their correlation with the dependent variables and the strong association with the main outcome (mortality).

Indeed, HDL-C, eGFR (kidney function) and TMAO were included as covariates in the table 2A because they significantly correlated with ABCA1/G1-mediated macrophage cholesterol efflux, whereas in table 2B, we included hs-cTnT, GRACE score, HDL-C and CRP.

In the case of Table 3A, we included eGFR and ABCA1/G1-mediated macrophage cholesterol efflux. For Table 3B, we included hs-cTnT, GRACE score and CRP as commented above.

LDL-C was not considered as a covariate since LDL particles are excluded in the APOB-depleted plasmas and, thus, they could not affect the APOB-depleted plasma capacity to stimulate cholesterol efflux and neither correlated with gut microbial-derived metabolites.

These points have now been clarified in the section 3.4.

3.4 ABCA1/G1-mediated macrophage cholesterol efflux is ~.

Is statistical analysis performed using only STEMI patients in this paragraph? The data for which group is difficult to understand. There is not enough explanation of the data thorough the manuscript.

Three groups were always included in the statistical analysis, except for GRACE 2.0 risk scores which were only determined in STEMI patients and, thus, included in the multivariate analyses. As requested, the 3.4 section has been rewritten for a better explanation of  the data.

Figure 3

There is a large variation in Figure 3A-C. The authors should increase the number. Please add the explanation of “macrophage cholesterol efflux (% at 24h)”.

In figure 3A-C, we originally aimed to evaluate the effects of increasing concentrations of TMAO, γBB, and TML on the expression of macrophage transporters in vitro. We did not find any consistent change with the different treatments albeit it is true that a low point number was included. However, gene expression often predicts poorly protein concentration and activity and the most relevant point for this paper was that gut microbial-derived metabolites did not affect the ability of APOB-depleted plasma to enhance the ABCA1/G1-mediated macrophage cholesterol efflux in vitro. For this reason, we have now only included these data in the figure 3. Also, we have now clarified the rationale of this experiment in section 3.5 and added an explanation in the figure 3 legend.

Round 2

Reviewer 2 Report

This manuscript was improved by revision.